# CoFe_2_O_4_@HaP as Magnetic Heterostructures for Sustainable Wastewater Treatment

**DOI:** 10.3390/ma16072594

**Published:** 2023-03-24

**Authors:** Raluca-Ștefania Dănilă, Ioan Dumitru, Maria Ignat, Aurel Pui

**Affiliations:** 1Faculty of Chemistry, University of Alexandru Ioan Cuza, Boulevard Carol I No. 11, 700506 Iasi, Romania; 2Faculty of Physics, University of Alexandru Ioan Cuza, Boulevard Carol I No. 11, 700506 Iasi, Romania

**Keywords:** magnetic nanoparticles, hydroxyapatite, adsorption

## Abstract

The aim of this study was to synthesize a CoFe_2_O_4_@HaP nanocomposite (HaP-Hydroxyapatite) through the coprecipitation method in aqueous solution, with the purpose of using it in adsorption processes for the removal of Congo Red dye from aqueous solutions. Fourier Transform Infrared Spectroscopy (FT-IR) was used to characterize the synthesized material, identifying absorption bands specific to the functional groups of cobalt ferrite (Fe-O and Co-O at 603 and 472 cm^−1^) and hydroxyapatite PO_4_^3−^ at 1035, 962, 603 and 565 cm^−1^. Powder X-ray diffraction confirmed the cubic spinel structure of cobalt ferrite (S.G Fd-3m) and the hexagonal structure of hydroxyapatite (S.G P63/m). The nanocomposite’s crystallite size was calculated to be 57.88 nm. Nitrogen adsorption/desorption isotherms and BET specific surface area measurements were used to monitor textural parameters, revealing an increase in specific BET surface area when cobalt ferrite nanoparticles (15 m^2^/g) were introduced into the hydroxyapatite heterostructure (34 m^2^/g). Magnetic properties were investigated by interpreting hysteresis curves in the ±10 kOe range, with the nanocomposite showing a saturation magnetization of 34.83 emu/g and a coercivity value of 0.03 kOe. The adsorption capacity of the CoFe_2_O_4_@HaP nanocomposite is up to 15.25 mg/g and the pseudo-second-order kinetic model (Type 1) fits the data with a high correlation coefficient of 0.9984, indicating that the chemical adsorption determines the rate-determining step of the process. The obtained nanocomposite is confirmed by the analyses, and the absorption measurements demonstrate that it can be utilized to degrade Congo Red dye.

## 1. Introduction

The adsorption technique is becoming one of the most effective alternatives for industrial water treatment, due to the fact that it presents a technology that can be used for the removal of wide classes of contaminants. This technique also occupies an important place in wastewater treatment processes to remove pollutants that are not easily biodegradable, and at the same time, through adsorption processes, the materials used can be recovered and regenerated. A general definition of adsorption could be expressed as a surface phenomenon where the substance to be retained from a solution (adsorbate) is adsorbed on the surface of a material called an adsorbent. The adsorption phenomenon can be of two types: physisorption, in which the adsorbate binds to the adsorbent due to van der Waals interactions, and chemisorption [1].

Due to the increase in the world population, the demand and need for clean and potable water have greatly increased. As a consequence of this demographic evolution, the amount of lethal chemicals such as heavy metals and pharmaceuticals, discharged into the sewers, has also greatly increased and at the same time can produce harmful effects on the environment [2,3]. The pollution of water with various dyes, used in the textile and chemical industries, has led to an increase in the interest of scientists to discover new and effective methods of eliminating this category of contaminants. For example, Birniwa and co-workers synthesized a polypyrrole-polyethyleneimine (PPy-PEI) adsorbent material with the aim of it being used for the removal of methylene blue dye from aqueous solutions [4]. Al-Mahbashi and co-workers studied a process to remove dyes used in batik production from industrial waters [5]. Another group of researchers, Al-dhawi and co-workers, studied the synthesis of a material that can extract boron from polluted waters [6].

Dyes can be defined as substances that, once applied to a substrate, impart color through a process that modifies the crystalline structure of the colored substances. Such substances with coloring potential are widely used in the textile, pharmaceutical, cosmetic industries, etc. The structures of dyes are complex and diverse. Currently, more than 5000 variants of dyes with different chemical structures are known (Dye Index) [7,8,9]. The molecular structures of dyes contain reactive groups that can form covalent bonds, for example with hydroxyl groups of cellulose fibers, or with amino groups of protein fibers, and thus become reactive dyes, making their elimination process more difficult [8]. One method of removing dyes from wastewater is the adsorption technique. In this study, the removal of Congo Red dye through the process of adsorption from an aqueous solution was pursued. The studied dye belongs to the class of anionic diazo dyes. As a structure, this compound contains a monoazo bond (R=N=R’). Congo Red dye presents four functional groups—two of the amino type; two of the sulfonic type—and various molecular structures that depend on the pH of the dye solution (for example, at neutral pH, the sulfonic groups form a polyanionic structure).

The IUPAC name of the dye is disodium disodium 4 amino-3-[4-[4-(1-amino-4-sulfonato-naphtalen-2-yl)diazenylphenyl]phenyl]diazenyl-naphtalene-1-sulfonate, with the general formula C_32_H_22_N_6_Na_2_O_6_S_2_. Congo Red dye belongs to the group of benzene compounds, presenting in its structure aromatic amines. As a result of the metabolism of the dye, benzidine is obtained, which produces harmful effects in living organisms [10]. The toxicity of these dyes is high in the case of their absorption and reaching the liver, because, following the interaction with azoreductases, intestinal microfluoride and bacteria on the surface of the skin can metabolize, and thus aromatic amines can be formed whose presence in the body is very harmful. Despite the risks resulting from the use of this dye, there are sectors in which it is still used, for example in the textile industry because of its affinity with cellulose fibers, and in the cosmetic, pharmaceutical, and stationery industries. Setyawati and co-workers [11] carried out a study in which they functionalized Congo Red dye to improve its photovoltaic performance (by forming complexes with metals such as Fe, Ni, and Co). The purpose of this functionalization was to use the complexes in solar cells. Another group of researchers, Yakupava and co-workers [12], opted for the application of the Congo Red dye in the identification of amyloid diseases in the liver, kidneys, brain, and other human or animal organs. As this dye is frequently used in various fields, the risk of its discharge into wastewater is increasing. In recent years, many options have been tried to remove the dye from water sources, and it is desirable to discover optimal methods for its removal both from the point of view of efficiency and from the point of view of cost.

Currently, nanomaterials are gaining significant importance in adsorption processes, combined with magnetic separation processes, due to their high capacity to remove pollutants from aqueous media [13]. In the framework of adsorption studies, the magnetic properties of nanomaterials have become increasingly researched. The main interest is related to the possibility of extracting adsorbents from solutions and their reuse. Kuncser V. and co-workers performed a study in which they tested the magnetic properties and thermal stability of polyvinylidene fluoride—Fe_2_O_3_ nanocomposites [14].

Nanocomposites such as MFe_2_O_4_ ferrites introduced into the hydroxyapatite structure are promising candidates for many applications of catalysis, adsorption, and use in industry in combating environmental pollution, etc. The increased biocompatibility of compounds from the apatite class led to the discovery of nanocomposites with applications in medicine. For example, Karthi and co-workers functionalized magnetite with fluorapatite to be used in biomedical applications [15]. Hydroxyapatite shows good sorption properties, which is why Biedrzycka and co-workers conducted a study in which they synthesized and characterized a HaP/Iron oxide type composite to be tested in medical and environmental protection applications [16].

The use of nanomaterials as adsorbents has become an increasingly fertile research area. Due to their reduced volume, nanomaterials thanks to their large surface area, containing a large number of free adsorbate sites, and high porosity, make these materials one of the strongest competitors on the market for contaminant extraction from various aqueous solutions [17]. Nanomaterials such as magnetic nanoparticles, carbon nanotubes, and porous materials are widely used as new adsorbents. The efficiency of adsorption processes can be improved by functionalizing nanomaterials, a procedure that leads to an increase in adsorption and the attraction between the adsorbent molecules and the molecules to be removed becoming greater [18,19,20,21].

Previous studies have been carried out on the synthesis of ferrites by various techniques. The experience gained regarding different ferrite synthesis methods led to the realization of the present study. Starting from ferrites, the synthesis of a nanocomposite composed of cobalt ferrite and hydroxyapatite was desired with the aim of being used in various applications.

The aim of the present study is to investigate the potential application of CoFe_2_O_4_@HaP nanocomposite for the removal of Congo Red dye from aqueous solutions. To achieve this goal, the study carries out the synthesis of the nanocomposite using the coprecipitation method, followed by the characterization of the material using complementary techniques such as Fourier Transform Infrared Spectroscopy (FT-IR), Powder X-ray diffraction, and magnetic measurements. Valuable insights into the suitability of the CoFe_2_O_4_@HaP nanocomposite for practical applications in water treatment were determined through adsorption studies by assessing the adsorption capacity of the CoFe_2_O_4_@HaP nanocomposite and by analyzing the kinetic models that govern the process.

## 2. Materials and Methods

### 2.1. Chemicals

The reagents used in this study for CoFe_2_O_4_ synthesis are: CoCl_2_•6H_2_O (ACS reagent, 99% Sigma-Aldrich, St. Louis, MO, USA), FeCl_2_•6H_2_O (Fisher Scientific, laboratory reagent grade, Waltham, MA, USA), NaOH (ROTH 98%, Carl Roth, Karlsruhe, Germany), carboxymethyl cellulose (CMC) (Sigma-Aldrich, St. Louis, MO, USA). The materials used in the synthesis of the CoFe_2_O_4_@HaP nanocomposite are: Ethanol (99.8% ROTH, Carl Roth, Karlsruhe, Germany), Hexadecyltrimethylammonium bromide (CTAB, Sigma Aldrich, St. Louis, MO, USA), H_3_PO_4_ (85% ROTH, Carl Roth, Karlsruhe, Germany), NH_4_OH (25% CHIM REACTIV SRL, Bucharest, Romania), Ca(NO_3_)_2_ (Sigma-Aldrich, St. Louis, MO, USA).

### 2.2. Synthesis

The nanocomposite of cobalt ferrite and hydroxyapatite (CoFe_2_O_4_@HaP 1:1 mass ratio between the magnetic component and the biomaterial) was obtained in two steps. The starting point was cobalt ferrite nanoparticles, which were synthesized by the co-precipitation method in aqueous solution, adapted from [22,23]. A solution of CoCl_2_•6H_2_O 0.2 M was mixed with a solution of FeCl_2_•6H_2_O 0.4 M under continuous magnetic stirring. To prevent particle agglomeration, a 1% carboxymethyl cellulose (CMC) solution was added as a surfactant. The precipitation of the oxides was done by adding a 3 M NaOH solution. The pH was checked and adjusted to reach a value between 11–12. The black precipitate obtained was washed with distilled water to remove NaCl resulting from the synthesis process. After the synthesis of cobalt ferrite nanoparticles, we proceeded to introduce them into the hydroxyapatite structure. Cobalt ferrite nanoparticles were dispersed in 25 mL ethanol, together with 0.1 g CTAB (surfactant agent) and 25 mL 0.3 M H_3_PO_4_ under continuous mechanical stirring at 60 °C. The pH of the mixture was corrected with NH_4_OH followed by dropwise addition of 0.5 M Ca(NO_3_)_2_ solution and allowed to age overnight. The particles were washed with distilled water until the pH reached a neutral value. Then they were separated from the solution by decantation, dried in an oven for 24 h at 70 °C, and calcined for 4 h at 500 °C, in a high-temperature furnace. The working protocol was modified according to [24].

### 2.3. Characterization Methods and Equipment

The structural properties of the cobalt ferrite and hydroxyapatite nanoparticles were investigated by complementary analytical techniques. Fourier Transform Infrared Spectroscopy (FT-IR) was performed using a Jasco 660 Plus FT-IR spectrophotometer (JASCO INTERNATIONAL Co., Ltd., Tokyo, Japan) via the pilling method. X-ray diffraction (XRD) was conducted using a Shimadzu LabX XRD-6000 difractometter (Shimadzu, Kyoto, Japan) with a copper radiation source (λ = 1.54 Å) in the 2θ range of 20° to 80° at a speed of 2°/min, and the data were processed with X՚Pert HighScore Plus software, version 2.2.3, which includes ICSD databases of analyzed compounds.

The textural properties of the nanocomposite were analyzed by nitrogen adsorption/desorption isotherms and BET specific surface area measurements using a Nova 2000e Quantachrome instrument (Quantachrome Instruments, Boynton Beach, FL, USA). The magnetic properties were measured using a vibration magnetometer in the ±10 kOe range at room temperature.

The general diagram of the stages covered in this study is presented in Figure 1.

The study has three main stages: the synthesis of the CoFe_2_O_4_@HaP nanocomposite, the characterization of the obtained material, and adsorption studies of Congo Red dye.

### 2.4. Dye Adsorption Experiments

Adsorption experiments with a focus on CoFe_2_O_4_-HaP dosage, contact time, and temperature were carried out in order to establish the ability of the material synthesized as a potential adsorbent for the treatment of wastewater contaminated with Congo Red dye. The general diagram of the Congo Red dye removal process is presented in Figure 2.

The adsorption studies were made using a UV-3100PC UV-Vis spectrophotometer (VWR International Europe bvba, Leuven, Belgium). The absorbance of the supernatant of each sample was measured at 497 nm. The following equation was used to calculate the adsorption capacity:(1)$$q_{e}=\frac{\left(C_{i}-C_{f}\right)V}{m}$$
where *q_e_* is the adsorption capacity (mg/g), *C_i_* and *C_f_* are the initial dye concentration and equilibrium dye concentration (mg/L), *V* is the volume of dye solution (L), and *m* is the quantity of CoFe_2_O_4_-HaP used (g).

## 3. Results

### 3.1. Powder X-ray Diffraction Analysis (XRD)

In order to determine the structure and phases of the synthesized nanocomposites, their analysis was carried out by X-ray powder diffraction. Diffractograms of the CoFe_2_O_4_, CoFe_2_O_4_@HaP, and Hydroxyapatite (HaP) are presented in Figure 3. The cobalt ferrite sample (CoFe_2_O_4_) corresponds to a single phase with a cubic structure belonging to the space group Fd-3m (ICSD 98-007-9524). For the sample of cobalt ferrite and hydroxyapatite, in addition to the characteristic structure of ferrite, a hexagonal structure specific to hydroxyapatite with space group P63/m was identified. Analysis of the hydroxyapatite nanoparticles again confirmed the corresponding hexagonal structure as well as their membership in the space group P63/m (ICSD 98-009-4268).

The crystallite sizes of the three samples were calculated using the Debye–Scherrer equation [25]:D = (K × λ)/(B × cos θ_B_);(2)D = crystallite size (nm);K = constant dependent on the shape of the crystallite (0.89) (dimensionless);λ = wavelength of X-rays (nm)B = FWDHM (Full Width at Half Maximum) (degree)θ_B_ = Bragg angle (degree).

The values of the crystallite size for the three samples are presented in Table 1.

Lattice parameter—a (Å), unit cell volume—V (Å^3^), and density—ρ_XRD_ (g/cm^3^) obtained from the interpretation of the diffractograms of the three samples are given in Table 2.

A comparison of the values in Table 2 with those from the ICSD database shows very good agreement. The CoFe_2_O_4_@Hap nanocomposite was successfully obtained.

### 3.2. FT-IR Spectroscopy

FT-IR spectra provide information about the valence or deformation vibrations of bonds such as metal-oxygen (M-O), and O-H. This is very useful information in establishing the composition/structure of chemical compounds. The FT-IR spectra for the samples of CoFe_2_O_4_, Hydroxyapatite (HaP), and CoFe_2_O_4_@HaP are shown in Figure 4.

In the spectrum of the cobalt ferrite sample, an absorption band was identified at 3432 cm^−1^ corresponding to the valence vibration of the O-H bond in the water molecules. At 1637 cm^−1^, the characteristic absorption band of the COO^−^ group from the unreacted surfactant molecules (CMC) was observed. The metal-oxygen, Fe-O and Co-O bonds, in the ferrite structure were found following the identification of the absorption bands at 603 and 472 cm^−1^ [26]. The infrared spectrum of the hydroxyapatite sample showed absorption bands at 3570 cm^−1^ and 3432 cm^−1^. These correspond to the deformation and valence vibrations of the OH group from hydroxyapatite, respectively. At 1646 cm^−1^ and 1416 cm^−1^ the absorption bands of the CO_3_^2−^ group were identified. The PO_4_^3−^ groups from the hydroxyapatite structure contributed the following absorption bands: 1035, 962, 603, and 565 cm^−1^ [27]. In the FT-IR spectrum of the CoFe_2_O_4_@HaP nanocomposite, the absorption bands corresponding to the Fe-O and Co-O bonds in the ferrite structure were found. Also, the presence of PO_4_^3−^ and OH groups from hydroxyapatite were identified in the spectrum of the nanocomposite. The results illustrate the successful synthesis of the CoFe_2_O_4_@HaP nanocomposite and provide insight into its structural properties. As mentioned above, the infrared spectra of the individual components, i.e., cobalt ferrite and hydroxyapatite, are indicated by the presence of characteristic absorption bands corresponding to their respective functional groups. These results suggest the potential use of the CoFe_2_O_4_@HaP nanocomposite in applications such as wastewater treatment through adsorption processes.

### 3.3. Magnetic Properties

To identify the magnetic properties, the synthesized samples were analyzed using the VSM (Vibrating Sample Magnetometer) technique, with a PMC MicroMag 3900 VSM equipment (PMC- Princeton Measurement Corporation, Princeton, Montgomery, NJ, USA) within ±10 kOe, at room temperature. The hysteresis loops as well as the magnetizations obtained for the two samples are shown in the graphs in Figure 5 (CoFe_2_O_4_) and Figure 6 (CoFe_2_O_4_@HaP).

Through the hysteresis loops shown in the graphs in Figure 5 and Figure 6, specific magnetic parameters such as saturation magnetization (M_s_), remanent magnetization (M_r_), coercivity (H_c_), and M_r_/M_s_ ratio could be identified and are presented in Table 3.

The low values of the M_r_/M_s_ ratios obtained for the two samples analyzed indicate the presence of fractions of superparamagnetic particles. Superparamagnetism usually occurs for ferrimagnetic or ferromagnetic nanoparticles made up of very small particles. After analyzing the hysteresis loops from Figure 5 and Figure 6, a decrease in the saturation magnetization of the nanocomposite was observed (Table 3). This may be caused by the mass ratio used in the synthesis process. As this was 1:1, we have in the composition of the material 50% cobalt ferrite and 50% hydroxyapatite. It is known that hydroxyapatite has no magnetic properties, so its presence at a percentage of 50% in the structure of the nanocomposite led to a decrease of approximately 50% in the saturation magnetization. There was also a decrease in coercivity for the nanocomposite. The reason for this decrease may be the fact that after the introduction of ferrite in the hydroxyapatite structure there was an increase in the crystallite size of the nanocomposite (Table 1), and from the literature, it was observed that an increase in coercivity with an increase in the size of the magnetic material until reaching the value of the critical diameter can be followed by a decrease in coercivity values [28,29,30].

### 3.4. BET Analysis (Brunauer–Emmett–Teller)

The textural properties of cobalt ferrite and hydroxyapatite were identified following the analysis of the nitrogen adsorption-desorption processes and the interpretation of the isotherms corresponding to the two processes (Figure 7(a(1),b(1),c(1))).

The graphs in Figure 7(a(1),b(1),c(1)) show the adsorption-desorption isotherms for CoFe_2_O_4_—a(1), Hydroxyapatite (HaP)—b(1) and CoFe_2_O_4_@HaP—c(1) samples. They show the appearance of a type IV isotherm, with a hysteresis of type H3 according to the IUPAC classification, with a capillary condensation at high relative pressures, characteristic of mesoporous materials with slit-type pores [31].

The textural parameters resulting from processing the adsorption-desorption isotherms for the synthesized samples are shown in Table 4.

As can be seen from the data in Table 4, the introduction of the cobalt ferrite nanoparticles into the hydroxyapatite matrix led to an increase in the BET specific surface area and a decrease in the total pore volume, while the pore diameter remained constant. Magnetic cobalt ferrite nanoparticles are prone to agglomerate, which is the reason for adding carboxymethylcellulose (CMC) in the synthesis process (Section 2. Materials and methods, Section 2.2. Synthesis). Thus, upon the addition of the surfactant solution (CMC 1%), the nanoparticles remained dispersed in the stabilized suspension. However, when they were separated and dried, the nanoparticles agglomerated and showed a low relative BET specific surface area. Therefore, in order to keep an increased specific surface area in the solid state, in the present study, the prepared cobalt ferrite nanoparticles were dispersed in a solid matrix of hydroxyapatite to avoid their agglomeration in the solid phase and to increase their contact surface. Therefore, the use of the surfactant cetrimonium bromide (CTAB) had a double role in the process of introducing cobalt ferrite nanoparticles into hydroxyapatite: firstly to disperse the nanoparticles in solution around which the hydroxyapatite will precipitate and a second role to creates porosity around magnetic cobalt ferrite nanoparticles. Thus, the addition of cobalt ferrite in the hydroxyapatite matrix led to the dispersion and not the agglomeration of the particles, and the value of the BET specific area for the nanocomposite, as expected, was found to be twice that of the pure nanoparticles.

### 3.5. Adsorption Studies

Adsorption studies were made using UV-Vis spectrophotometry.

In this study, three experimental factors were investigated: CoFe_2_O_4_-HaP dosage, contact time, and temperature. The adsorption results are depicted in Figure 8.

One of the experimental factors of the adsorption process that has an impact on the adsorption capacity of the adsorbent is the quantity of the material used. Thereby, in this study, five CoFe_2_O_4_-HaP dosages were investigated (Figure 8a). A CoFe_2_O_4_-HaP dosage of 0.47 g/L leads to an adsorption capacity of 14.9 mg/g. By increasing the dosage to 0.49 g/L, a slight increase in the adsorption capacity is observed to 15.25 mg/g. At the same time, the results show that higher dosages of CoFe_2_O_4_-HaP adsorbent cause lower adsorption capacities—0.96 g/L, 1.9 g/L, and 3.04 g/L CoFe_2_O_4_-HaP dosages led to adsorption capacities of 13.07 mg/g, 12.58 mg/g, and 9.15 mg/g, respectively. Another adsorbent reported previously in the literature presented the same behavior. Zhao and co-workers explained this trend by that higher adsorbent dosages cause the adsorption sites to overlap and not be effectively exploited [32]. Based on the results obtained, the subsequent CoFe_2_O_4_-HaP dosage used for the effect of the other two adsorption factors, contact time and temperature, was 0.49 g/L.

Figure 8b compares the adsorption capacities of CoFe_2_O_4_-HaP material at various time intervals, from 5 min to 120 min, using an initial dye concentration of 30 mg/L. According to the data obtained, the Congo Red dye is adsorbed rapidly in the first 5 min of contact time: 10.83 mg/g, which represents ~71% of the total adsorption capacity of the CoFe_2_O_4_-HaP adsorbent. The presence of numerous active sites on the surface of the adsorbent could explain the fast adsorption [33,34,35]. After 8 min of contact time, the CoFe_2_O_4_-HaP nanocomposite has the ability to adsorb 12.07 mg/g of dye. The adsorption capacity slightly increases from 15.14 mg/g to 15.24 mg/g between 10 min and 50 min of contact time. After 60 min and 120 min of contact time, the adsorption capacity is unchanged. So, ~50 min of contact time can be considered sufficient in order to achieve the maximum adsorption capacity. The literature shows an adsorbent synthesized based on hydroxyapatite nanoparticles that shows the same trend: the adsorption process occurs quickly (5–10 min) [36]. In addition, Sirajudheen and co-workers prepared an adsorbent based on chitosan-supported graphene oxide-hydroxyapatite composite and used it to clean water sources polluted with different dye contaminants. Their results show that the adsorption process is very fast, with a contact time of 40 min being enough to attain the equilibrium [37].

A brief comparison of the data obtained in this study versus other materials used for Congo Red dye reported in the literature is further discussed. It is recommended to take into consideration that the value of adsorption capacity strongly depends on the type of material used and the experimental adsorption conditions. AgNPs-functionalized-HaP material was synthesized by the research group of Azeez and co-workers. The investigation showed that using an adsorbent dosage of 0.5 g/50 mL and an initial concentration of 50 mg/L, the material has an ability to adsorb 24.41 mg/g of dye molecules [38].

Nguyen and co-workers proposed three adsorbents, abbreviated as MIL–53(Al), CoFe_2_O_4_, and CoFe_2_O_4_@MIL–53(Al), in order to adsorb Congo Red dye [39]. The experimental conditions proposed by the authors: 0.01 g/20 mL solution, 10 mg/L initial dye concentrations demonstrate the following adsorption capacities: 8.01 mg/g, 15.62 mg/g, and 19.61 mg/g. The results showed that the materials followed the order: MIL–53(Al) < CoFe_2_O_4_ < CoFe_2_O_4_@MIL–53(Al).

Simonescu and co-workers synthesized CoFe_2_O_4_ nanoparticles and CoFe_2_O_4_-Chitosan composite material and tested them for Congo Red dye adsorption. In order to establish how the contact time affects the adsorption capacity, the samples were collected at different time intervals until 360 min. Using an adsorbent dosage of 0.01 g/25 mL Congo Red dye solution of 102.81 mg/L initial concentration, the results showed that the adsorption capacities are 16.18 mg/g for CoFe_2_O_4_-Chitosan adsorbent and 146.5 mg/g for CoFe_2_O_4_ adsorbent [40].

The data obtained from the experiment regarding the effect of contact time were fitted using different kinetic models: pseudo-first-order, pseudo-second-order (Type 1, Type 2, Type 3, and Type 4), and intraparticle diffusion. The plots are shown in Figure 8c–h and the kinetic parameters for the investigated models calculated from the slope and intercept using the equations presented in the literature [41,42] are listed in Table 5.

From Table 5 it can be seen that according to the pseudo-first-order kinetic model, the correlation coefficient, R^2^, has a lower value. The pseudo-first-order rate, $k_{1}$, calculated from the slope is 0.0672 min^−1^.

The R^2^ determined using the pseudo-second-order kinetic model, Type 1 is 0.9996 while pseudo-second-order kinetic model Type 2, pseudo-second-order kinetic model, Type 3, and pseudo-second-order kinetic model, Type 4 show values of R^2^ below 0.85. Moreover, the experimental adsorption capacity (q_e_) and the calculated adsorption capacity value (q_e_ cal) for pseudo-second-order, Type 1 have close values, 15.25 mg/g vs. 15.45 mg/g. The pseudo-second-order rate, $k_{2}$, calculated for the other three types of pseudo-second-order model are 0.0280 g/mg min, 0.0434 g/mg min, and 0.0218 g/mg min. The q_e_ cal for pseudo-second-order kinetic models, Type 2–Type 4, are 16.13 mg/g, 15.92 mg/g, and 16.59 mg/g, respectively.

By plotting *q_t_* vs. *t*^1/2^ (Figure 8h) it was found that the adsorption process of Congo Red dye using CoFe_2_O_4_-HaP adsorbent takes place in two stages. In addition, the intraparticle diffusion model does not fit the adsorption data, with the R^2^ values corresponding for both stages being lower.

Therefore, it can be highlighted that the rate-determining step of the adsorption process for Congo Red using CoFe_2_O_4_-HaP adsorbent is determined by chemical adsorption [43,44].

Figure 8i contains the results of the impact of the temperature of the dye solution on the CoFe_2_O_4_-HaP nanocomposite’s ability to adsorb the dye. The working temperatures were 296 K, 307 K, and 319 K using an initial dye concentration of 30 mg/L and 0.49 g/L CoFe_2_O_4_-HaP dosage. In addition, the effect of temperature was studied with intermittent stirring at natural pH. The results show that an increase in temperature of 11 K leads to an increase in adsorption capacity from 15.25 mg/g to 20.17 mg/g, while at a temperature of 319 K, the CoFe_2_O_4_-HaP adsorbent has an adsorption capacity of 27.97 mg/g.

The type of the adsorption process was established based on Gibbs free energy (ΔG°), entropy change (ΔS°), and enthalpy change (ΔH°) thermodynamic parameters, according to Equations (3) and (4) [38], and the results are list in Table 6. The ΔH° and ΔS° parameters were determined from the slope and intercept plot of *lnK_D_* vs. 1/*T*.
(3)$$lnK_{D}=\frac{\Delta S^{{}^{\circ}}}{R}-\frac{\Delta H^{{}^{\circ}}}{RT}$$
(4)$$\Delta G=-RTlnK_{D}$$

The ΔH° parameter showed that the adsorption process is endothermic. The negative values of ΔG° obtained for all three temperatures indicate the spontaneity and feasibility of the adsorption process. The affinity of the CoFe_2_O_4_-HaP adsorbent for Congo Red dye adsorption is highlighted by the positive values of ∆S°. The data are in agreement with the study performed by Babakir and co-workers [45].

The adsorption isotherm study was predicted using Langmuir and Freundlich isotherms (Figure 9).

By evaluating R^2^, the correlation coefficient, of both models, it was observed that the Freundlich model fits the adsorption data with an R^2^ value of 0.9844 compared with the Langmuir model (R^2^ = 0.8916). The Freundlich constant, K_F_, is 2.7 L/g and n has a value of 0.28.

The results obtained suggest that the nanocomposite synthesized in this study can be employed as a potential adsorbent to adsorb the Congo Red dye.

### 3.6. Evaluation of Reuse of CoFe_2_O_4_@HaP Composite

In order to establish the capacity of CoFe_2_O_4_@HaP composite to be re-used, cycles of adsorption/desorption were performed. For the desorption study, a solution of 0.1 M HNO_3_ was used. The results obtained (Figure 10) show that the CoFe_2_O_4_@HaP composite can be re-used as an adsorbent for the dye proposed, the adsorption capacity being 10.22 mg/g after three cycles.

## 4. Conclusions

The synthesized material exhibits great potential as an adsorbent for the removal of Congo Red dye, as demonstrated by the adsorption experimental results. The adsorption capacity is affected by the dosage of CoFe_2_O_4_-HaP, with higher dosages resulting in a decreased capacity. Moreover, increasing the temperature from 296 K to 319 K enhances the adsorption capacity from 15.25 mg/g to 27.97 mg/g. Equilibrium is achieved after approximately 50 min of contact time, with a rapid adsorption rate in the first few minutes. The pseudo-second-order kinetic model (Type 1) fits the data with a high correlation coefficient of 0.9984, indicating that chemical adsorption determines the rate-determining step of the process. Furthermore, the thermodynamic analysis suggests that the dye adsorption process is feasible, spontaneous, and endothermic in nature. Further research could include exploring the effectiveness of the CoFe_2_O_4_@HaP nanocomposite in the removal of other types of dyes or contaminants from wastewater, optimizing the synthesis process to improve the properties of the nanocomposite, and investigating the feasibility of scaling up the production of the nanocomposite for practical applications in water treatment. Furthermore, investigations on the photodegradation performance of the CoFe_2_O_4_@HaP nanocomposite for dye removal under visible light irradiation could be considered, in the expectation that such contributions will demonstrate the potential application of CoFe_2_O_4_@HaP nanocomposite as a highly effective adsorbent and photocatalyst for the removal of harmful dyes from wastewater.

## Figures and Tables

**Figure 1 materials-16-02594-f001:**
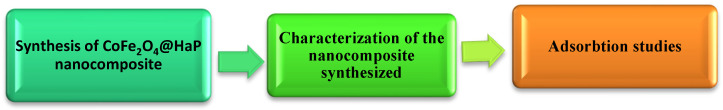
The general diagram of the stages covered in this study.

**Figure 2 materials-16-02594-f002:**
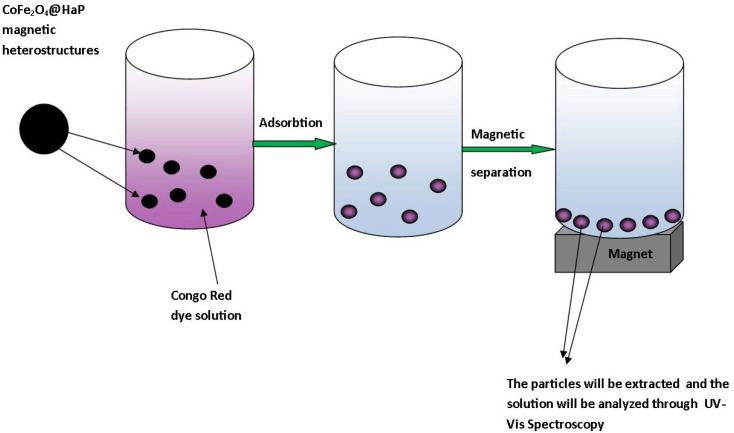
The general diagram of the Congo Red dye removal process.

**Figure 3 materials-16-02594-f003:**
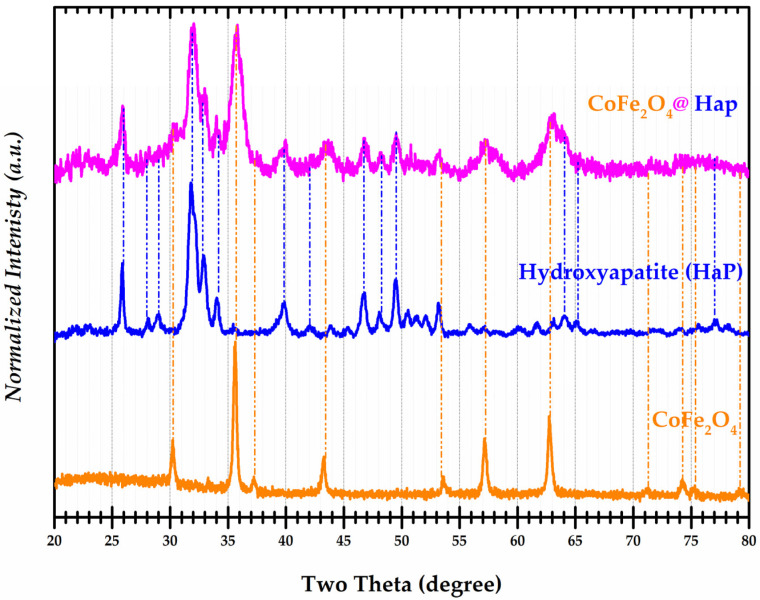
Normalized X-ray diffractograms for samples of Hydroxyapatite (HaP), CoFe_2_O_4_@HaP, and CoFe_2_O_4_.

**Figure 4 materials-16-02594-f004:**
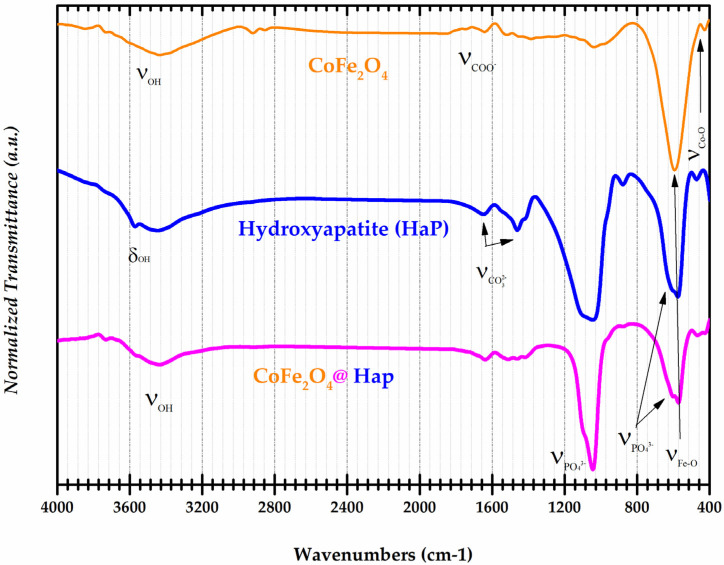
Normalized FT-IR spectra of CoFe_2_O_4_, Hydroxyapatite (HaP), and CoFe_2_O_4_@HaP samples.

**Figure 5 materials-16-02594-f005:**
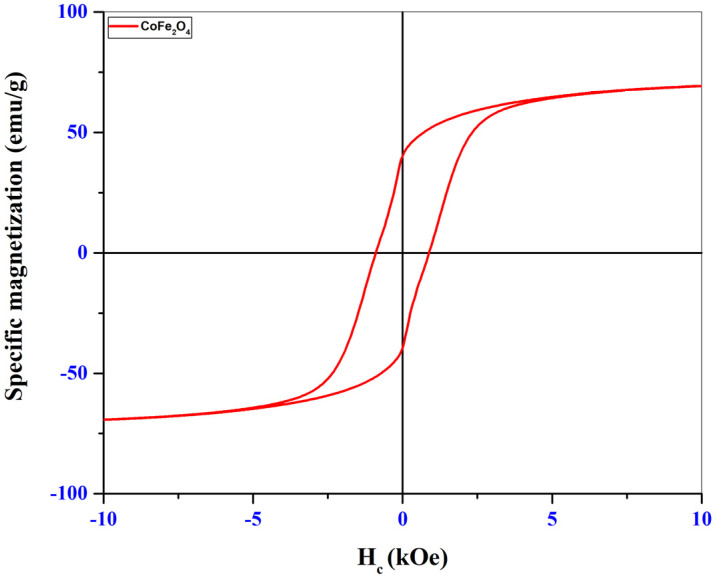
Magnetization of CoFe_2_O_4_ sample.

**Figure 6 materials-16-02594-f006:**
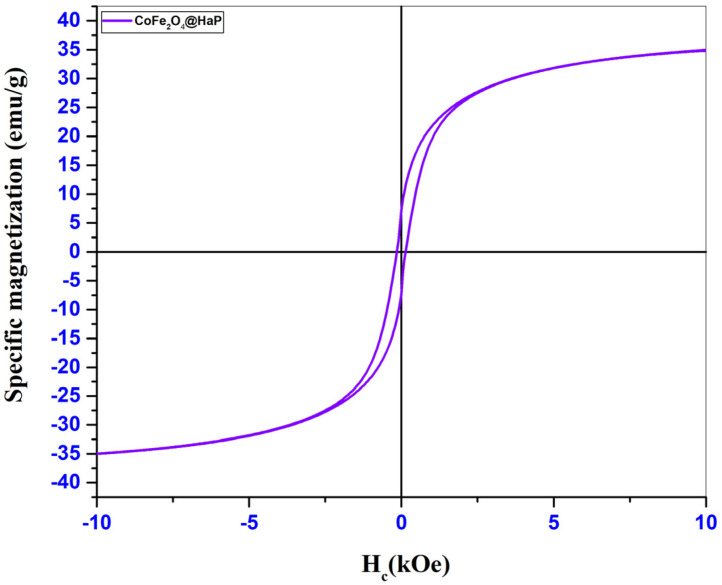
Magnetization of CoFe_2_O_4_@HaP sample.

**Figure 7 materials-16-02594-f007:**
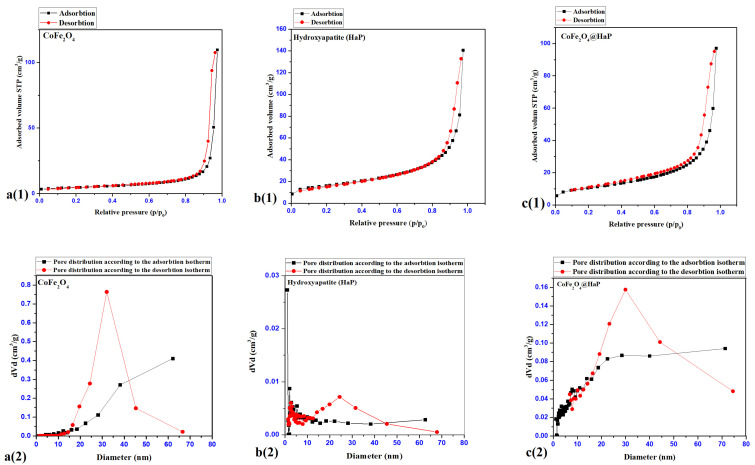
Adsorption-desorption isotherms (CoFe_2_O_4_—**a**(**1**), Hydroxyapatite (HaP)—**b**(**1**), CoFe_2_O_4_@HaP—**c**(**1**)), pore distributions after adsorption-desorption isotherms (**a**(**2**)–**c**(**2**)).

**Figure 8 materials-16-02594-f008:**
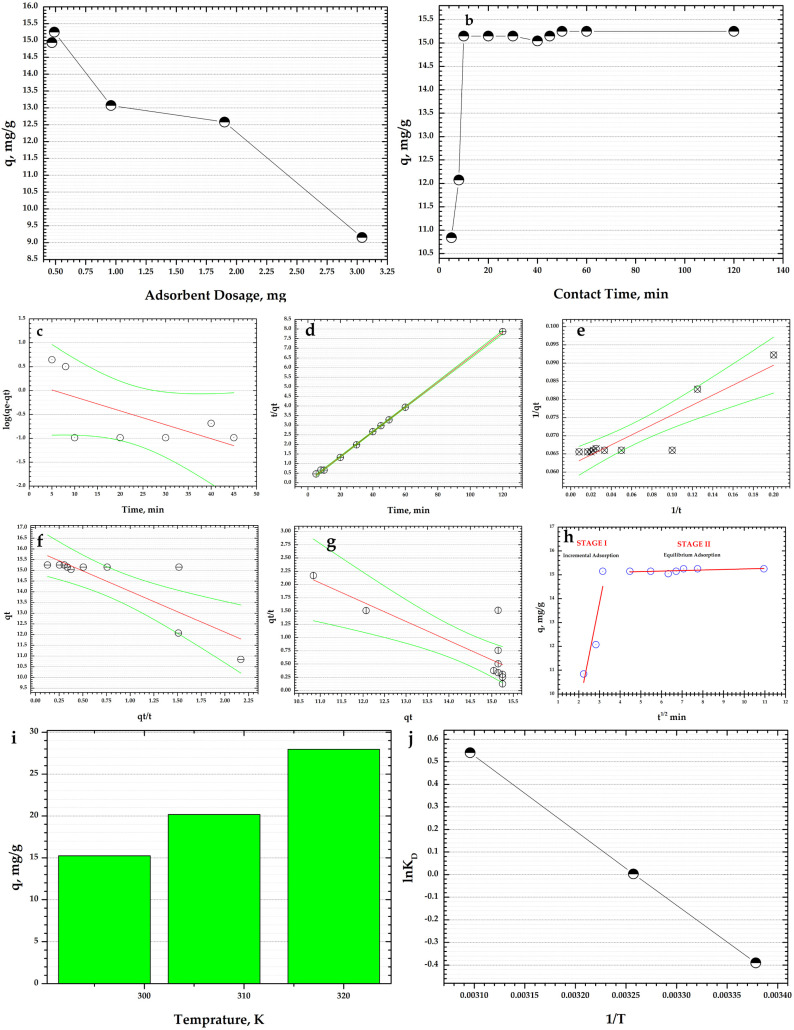
(**a**) Effect of CoFe_2_O_4_-HaP dosages on the dye adsorption (conditions: intermittent stirring at natural pH; room temperature, ~23 °C; initial dye concentration: 30 mg/L; contact time: 24 h); (**b**) Effect of contact time on the dye adsorption (conditions: intermittent stirring at natural pH; room temperature, ~23 °C; CoFe_2_O_4_-HaP dosage: 9.8 mg/20 mL solution, initial dye concentration: 30 mg/L); (**c**) pseudo-first-order kinetic model plot; (**d**) pseudo-second-order kinetic model plot, Type 1; (**e**) pseudo-second-order kinetic model plot, Type 2; (**f**) pseudo-second-order kinetic model plot, Type 3; (**g**) pseudo-second-order kinetic model plot, Type 4; (**h**) Intraparticle diffusion model plot; (**i**) Effect of temperature on the dye adsorption (conditions: intermittent stirring at natural pH; CoFe_2_O_4_-HaP dosage: 9.8 mg/20 mL solution, initial dye concentration: 30 mg/L; contact time: 50 min); (**j**) Van’t Hoff plot of the adsorption of dye on CoFe_2_O_4_-HaP adsorbent.

**Figure 9 materials-16-02594-f009:**
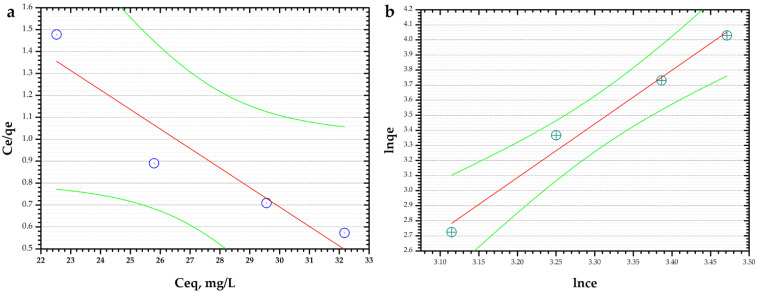
(**a**) Langmuir isotherm model; (**b**) Freundlich isotherm model.

**Figure 10 materials-16-02594-f010:**
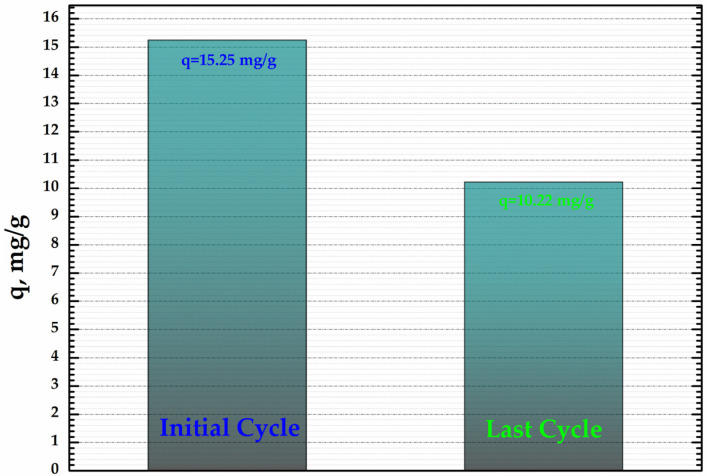
Direct comparison of adsorption capacity (q, mg/g) related to the initial cycle vs. the last cycle.

**Table 1 materials-16-02594-t001:** The values of the crystallite size for the three samples.

No.	Sample	Crystallite Size (nm)
1	CoFe_2_O_4_	44.46
2	Hydroxyapatite (HaP)	38.19
3	CoFe_2_O_4_@HaP	57.88

**Table 2 materials-16-02594-t002:** Experimental data obtained from the interpretation of the diffractograms of the three synthesized samples.

No.	Sample	a (Å)	V (Å^3^)	ρ_XRD_ (g/cm^3^)	Database
1	Hydroxyapatite (HaP)	9.4350	529.24	3.00	ICSD [98-009-4268]
2	CoFe_2_O_4_@HaP	8.3860	589.75	5.29	ICSD [98-007-9523]
3	CoFe_2_O_4_	8.3810	588.69	5.29	ICSD [98-007-9524]

**Table 3 materials-16-02594-t003:** Magnetic parameters for CoFe_2_O_4_ and CoFe_2_O_4_@HaP samples.

No.	Sample	M_r_ (emu/g)	M_s_ (emu/g)	H_c_ (kOe)	M_r_/M_s_
1	CoFe_2_O_4_	40.26	69.22	0.1	0.58
2	CoFe_2_O_4_@HaP	7.26	34.83	0.03	0.2

**Table 4 materials-16-02594-t004:** Textural properties of the synthesized nanocomposites.

No.	Sample	Specific Surface Area (m^2^/g) S_BET_	Total Pore Volume (cm^3^/g)	Pore Diameter (nm)
1	CoFe_2_O_4_	15	0.17	9
2	Hydroxyapatite (HaP)	55	0.12	9
3	CoFe_2_O_4_@HaP	34	0.07	9

**Table 5 materials-16-02594-t005:** Kinetic parameters of the investigated models.

Kinetic Model	Equation	Results
Pseudo-first-order kinetic model	$\log\left(q_{e}-q_{t}\right)=\log q_{e}-\frac{\left(k_{1}t\right)}{2.303}$	$k_{1}$ = 0.0672 R^2^ = 0.399
Pseudo-second-order kinetic model, Type 1	$\frac{t}{q_{t}}=\frac{1}{k_{2}q_{e}^{2}}+\frac{t}{q_{e}}$	q_e_ cal = 15.45 $k_{2}$ = 0.0725 R^2^ = 0.9996
Pseudo-second-order kinetic model, Type 2	$\frac{1}{q_{t}}=\left(\frac{1}{k_{2}q_{e}^{2}}\right)\frac{1}{t}+\frac{1}{q_{e}}$	q_e_ cal = 16.13 $k_{2}$ = 0.0280 R^2^ = 0.8308
Pseudo-second-order kinetic model, Type 3	$q_{t}=q_{e}-\left(\frac{1}{k_{2}q_{e}^{2}}\right)\frac{q_{t}}{t}$	q_e_ cal = 15.92 $k_{2}$ = 0.0434 R^2^ = 0.6829
Pseudo-second-order kinetic model, Type 4	$\frac{q_{t}}{t}=k_{2}q_{e}^{2}-k_{2}q_{e}^{2}q_{t}$	q_e_ cal = 16.59 $k_{2}$ = 0.0218 R^2^ = 0.6914
Intraparticle diffusion model, Stage 1	$q_{t}=K_{ID}t^{0.5}+C$	Stage 1: $K_{ID\;}$= 4.352 C = 0.7516 R^2^ = 0.8467
		Stage 2: $K_{ID\;}$= 0.0222 C = 15.022 R^2^ = 0.349

**Table 6 materials-16-02594-t006:** Thermodynamics parameters for Congo Red dye adsorption.

∆G° (kJ·mol^−1^)			∆H° (kJ·mol^−1^)	∆S° (kJ·mol^−1^·K^−1^)
**296 K**	**307 K**	**319 K**		
−30.71	−31.85	−33.09	31.76	103.86

## Data Availability

Data avaible upon request.

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
