# Peer review of "CoFe2O4@HaP as Magnetic Heterostructures for Sustainable Wastewater Treatment"

_materials, 2023, doi:10.3390/ma16072594_

Round 1
Reviewer 1 Report
The article presents the composite based on the Co-doped iron oxide coated with HAp as an effective adsorbent for Congo Red removal. It shows interesting results but needs some improvement.
The following comments need to be addressed:
1) The abstract should be reorganized to show the key experiments and the important findings.
2) Please normalize XRD patterns. For now, the negative counts appear on the y axis. I assume authors just overlapped the patterns. The units in the legend are missing. Please add them e.g. line 130 - D=crystallite size [unit should be added in bracket], etc.
3) Please describe the mass ratio of the core and shell. It is not clear what is the mass of the magnetic core and the shell at the moment.
4) The magnetometry results should be repeated, especially for the composite. Authors present almost the same values for CoFe2O4 (34.81 emu g-1)and CoFe2O4@HAp (34.65 emu g-1) - so the difference is just 0.16 emu g-1. This comment is related with the previous about the mass. The Ms values are very similar. Moreover, the coercivity is broader for bare magnetic particles what need an explanation - please describe it and compare the results with the litearture.
5) Please discuss the BET results. As the surface of composite is lower than for porous HAp, the role of the magnetic core should be highlighted.
6) Units in equations described in the adsorption studies are missing (concentration, etc.)
7) In the Introduction, authors mention the physisorption and chemisorption, while the adsorption mechanism studies are missing in the experimenatal work. Please present the isotherms, so the the most suitable model can be proposed.
8) It is not clear if authors reached the equilibrium. The measurements should be presented in longer time.
9) It is not clear why authors decided to dope iron oxide with cobalt. As it can change the optical properties, I would suggest to measure the band gap and propose photodegradation studies.
10) Authors propose Congo red as a model dye without discussion of the funtional groups interactions. It needs deeper discussion.
11) Intruduction is not sufficiently described. References in the introduction and the experimental part of the manuscript should be updated using the following papers:
https://doi.org/10.1557/jmr.2019.375
https://doi.org/10.1016/j.matchemphys.2017.02.047
https://doi.org/10.3390/ma15031139
12) Please correct some symbols e.g. 0C should have degree symbol, not zero in upper case; lower cases are missing e.g. Ms, Mr,
13) Please add discussion about reuse of proposed composite
14) Please add the results in function of pH and ionic strength.
Despite many comments I believe authors can add the results and impove the discussion the manuscript.
Reviewer 2 Report
The following report is based on my review of the manuscript entitled “CoFe2O4@HaP as magnetic heterostructures for sustainable Wastewater Treatment.” The manuscript materials-2253921 suits the scope of “Materials” and is also enjoyable. However, the following minor comments have been pointed out and need to be addressed properly to further improve the manuscript. They are as follows:
1. The present work is highly appreciable. However, The authors should add facts and figures in the abstract.
2. The introduction is well written. However, improvements can be made to enrich the manuscript to attract readers across the globe. The authors may refer to the following studies to improve the impact and visibility of the study.
· https://doi.org/10.3390/polym14163362
· https://doi.org/10.3390/su14116484
· https://doi.org/10.3390/pr11020453
3. The problem statement should be clearly stated.
4. Methodology is well written. However, the authors are suggested to add the flow chart of the study conducted. As well as subsections of the research conducted.
5. The results are stated very well in every subheading. However, the authors should move the equations and its description to section 2 (Materials and Methods)
6. The discussions lack support from previous studies. The authors should improve the discussions by supporting them with previous studies.
7. The conclusion section is also well written. However, the authors can add future recommendations.
8. Authors should avoid citations of non-index articles. Kindly, update the reference list with valid indexed citations. Authors should cite only recent 5 years' published articles. The journal referencing style should be followed throughout the manuscript.
